# Multi-Space Feature Fusion and Entropy-Based Metrics for Underwater Image Quality Assessment

**DOI:** 10.3390/e27020173

**Published:** 2025-02-06

**Authors:** Baozhen Du, Hongwei Ying, Jiahao Zhang, Qunxin Chen

**Affiliations:** 1School of Artificial Intelligence, Ningbo Polytechnic, Ningbo 315800, China; sunset16688@gmail.com; 2School of Electronic and Information Engineering, Ningbo University of Technology, Ningbo 315211, China; yhw@nbut.edu.cn; 3Ningbo Institute of Materials Technology and Engineering, CAS, Ningbo 315201, China; zhangjiahao@nimte.ac.cn

**Keywords:** underwater image quality assessment, underwater image, color space, multi-space feature, entropy

## Abstract

In marine remote sensing, underwater images play an indispensable role in ocean exploration, owing to their richness in information and intuitiveness. However, underwater images often encounter issues such as color shifts, loss of detail, and reduced clarity, leading to the decline of image quality. Therefore, it is critical to study precise and efficient methods for assessing underwater image quality. A no-reference multi-space feature fusion and entropy-based metrics for underwater image quality assessment (MFEM-UIQA) are proposed in this paper. Considering the color shifts of underwater images, the chrominance difference map is created from the chrominance space and statistical features are extracted. Moreover, considering the information representation capability of entropy, entropy-based multi-channel mutual information features are extracted to further characterize chrominance features. For the luminance space features, contrast features from luminance images based on gamma correction and luminance uniformity features are extracted. In addition, logarithmic Gabor filtering is applied to the luminance space images for subband decomposition and entropy-based mutual information of subbands is captured. Furthermore, underwater image noise features, multi-channel dispersion information, and visibility features are extracted to jointly represent the perceptual features. The experiments demonstrate that the proposed MFEM-UIQA surpasses the state-of-the-art methods.

## 1. Introduction

Underwater images, as an important ocean information medium, play an essential role in marine remote sensing because of their rich information and intuitiveness, which provide valuable data support for underwater navigation, marine geomorphological surveys, exploration, and other marine research [1]. However, due to the varying attenuation of different wavelengths of light in water, as well as the phenomena of water blur and scattering, underwater images often suffer from color shifts, hue deviations, and a decline in clarity, leading to significant image quality degradation [2]. These issues not only impact the visual quality of the images but also hinder image analysis and applications [3]. Consequently, the development of accurate and efficient underwater image quality assessment (UIQA) techniques is of paramount importance to the advancement of marine remote sensing and remote sensing research [4].

Image quality assessment (IQA) is a critical component in the field of image signal processing, typically divided into two major categories: subjective and objective assessments [5]. Objective evaluation methods, contingent upon the existence of a reference image, are further classified into full-reference IQA, reduced-reference IQA, and no-reference IQA (NR-IQA) [6]. Due to the unique characteristics of underwater environments, it is often challenging to obtain undistorted, high-resolution underwater images. Since NR-IQA methods do not rely on the original high-definition images, they are particularly practical and essential for the analysis of underwater image quality [7].

In recent years, there have been many studies on NR-IQA in atmospheric environments [8,9,10,11,12,13,14,15,16,17,18]. These methods, such as BRISQUE [8], NIQE [9], IL-NIQE [10], BPRI [11], DBCP [14], and VDA-DQA [15], have demonstrated promising results. However, these methods cannot be directly applied to the UIQA. The primary reason is the discrepancy between atmospheric imaging environments and underwater imaging environments. The attenuation of different wavelengths in water varies, leading to more severe color cast issues in underwater imaging. Additionally, due to the forward and backward scattering of light in water, issues such as insufficient brightness, reduced brightness uniformity, loss of detail, and decreased clarity are more serious, making the aforementioned methods unsuitable for direct application to underwater images. Consequently, it is essential to investigate the special quality assessment methods designed for underwater images.

To accurately assess the quality of underwater images, several SVR-based studies have been conducted [19,20,21,22,23,24,25,26,27,28], such as UCIQE [19], UIQM [20], CCF [21], and FDUM [22]. These methods primarily emphasize colorfulness, contrast, and sharpness, while incorporating certain characteristics of the human visual system (HVS) and mapping the quality of underwater images through nonlinear combinations. However, the effectiveness of the above methods in underwater image feature extraction still needs to be improved, and image entropy features are not effectively used to represent image quality, thus limiting their generalization capabilities. Deep learning-based IQA methods have achieved promising results [29,30] due to their powerful feature extraction capabilities. However, compared with the traditional IQA methods, deep learning IQA methods come at the cost of a significantly larger number of parameters, much longer training durations, and higher hardware configuration requirements. Moreover, unlike pre-selected artificial features, features automatically extracted by deep learning do not have explicit physical meanings. Therefore, UIQA methods that involve manual feature extraction remain the faster and more efficient approaches.

In this paper, a novel NR-UIQA approach MFEM-UIQA is proposed. Compared with the SVR-based UIQA method mentioned above, the proposed method can effectively extract underwater image features from multiple Spaces. In addition, considering that entropy is an important index to measure image information distortion and statistical properties, image entropy information features are used to represent image quality. Specifically, considering the varying attenuation of different wavelengths of light underwater, which leads to color cast, the underwater images are first subjected to spatial transformation from the RGB color space to the Opponent-Color (OC) space [31]. Then, the underwater chrominance difference (UCD) map is extracted based on the chrominance components of the OC space. Subsequently, statistical features are extracted from the UCD map. Moreover, the parameters of the Rayleigh distribution and entropy-based multi-channel mutual information (MI) features are extracted to further characterize chrominance features. Secondly, considering the underwater luminance, non-uniformity, and loss of image information, caused by the light scattering and absorption, the contrast features, luminance uniformity features, and entropy-based MI features based on luminance subband decomposition are extracted. Thirdly, taking into account underwater image noise and water blur, the noise feature, dispersion degree feature, and visibility feature are extracted to further characterize the perceptual features of the images.

The contributions of the proposed MFEM-UIQA are summarized as follows:(1)Considering the color cast of underwater images, the UCD map for underwater images is constructed based on the OC chrominance components, and statistical features are extracted. Moreover, the parameters of the Rayleigh distribution feature and entropy-based multi-channel MI features are extracted to characterize chrominance features;(2)Considering the underwater brightness attenuation, non-uniformity, and the loss of image detail information, the contrast features based on the gamma correction and the luminance uniformity features are extracted. Because of the high sensitivity of brightness features to image quality, logarithmic Gabor filtering is applied to the luminance space images for subband decomposition, and entropy-based MI features are extracted to characterize image luminance features comprehensively;(3)Considering the noise in underwater images and water blur, the noise feature is extracted by analyzing statistical features of the low-pass filtered luminance images. Additionally, the multi-channel dispersion degree feature and visibility feature based on multi-channel weighted enhanced metric estimation (EME) are extracted. The above features are combined as perceptual features to represent image quality.

The organization of the remaining content in this paper is as follows. Section 2 provides an overview of the related work. Section 3 details the proposed methodology. Section 4 presents and analyzes the experimental results. Finally, Section 5 concludes the paper with a summary and outlines potential directions for future research.

## 2. Related Work

### 2.1. Famous IQA Method

In this section, we provide a brief review of some Natural Scene Statistics-based (NSS) NR-IQA methods for atmospheric environments and natural scenes to accurately simulate human visual perception. Based on NSS, Mittal et al. [8] proposed a NR-IQA model BRISQUE, which measures the loss of “naturalness” in images by analyzing the statistical properties of local normalized luminance coefficients, laying the groundwork for subsequent studies. Mittal et al. [9] further proposed the natural Image Quality Evaluator (NIQE), which relies solely on the deviation from natural image statistics without any prior knowledge of distorted images or human subjective scores. Zhang et al. [10] presented an enhanced image quality assessment framework (IL-NIQE) that integrates natural image statistical features extracted from various clues, learning a multivariate Gaussian model from a set of pristine natural images to measure the quality of image patches and obtaining an overall quality score through average pooling. Min et al. [11] proposed a pseudo-reference image-based NR-IQA (BPRI), constructing pseudo-reference images and developing metrics for estimating blockiness, blurriness, and noise by calculating the structural similarity between the pseudo-reference and distorted images. Wang et al. [12] proposed an effective general NR-IQA model that reflects the degree of distortion caused by different image distortions based on the marginal distribution changes in the relative order coefficient. Based on the texture and structural information of images, Rajevenceltha et al. [13] proposed an effective NR-IQA metric to extract features based on local binary patterns and performed quality prediction through support vector regression (SVR). Considering the application of atmospheric haze image analysis, various NR-IQA methods for evaluating hazy and dehazed images have received widespread attention. [14,15,16,17,18]. Chu et al. [14] proposed a dehazed NR-IQA strategy DBCP based on the dark channel, bright channel priors, and contrast variations to indicate the level of haze. Guan et al. [15] introduced a NR-IQA method VDA-DQA for dehazed images based on the complex contourlet transform, visibility-aware features, and distortion-aware features. However, there are still limitations to applying the above approach directly to UIQA. This is mainly because underwater images are different from atmospheric images. They have unique challenges, such as uneven illumination and wavelength attenuation, which are not fully taken into account by these methods, making them unsuitable for direct application to the UIQA.

### 2.2. Underwater-Specific IQA

In recent years, researchers have achieved notable progress in the field of UIQA. Yang et al. [19] proposed the underwater color image quality evaluation (UCIQE) metric, which is based on the statistical distribution in the CIELab color space, utilizing a linear combination of chrominance, saturation, and contrast features to quantify the non-uniform color bias, blurriness, and low contrast issues in underwater images. Panetta et al. [20] introduced a novel NR-UIQA metric UIQM, which integrates colorfulness, sharpness, and contrast measurements, each considering the characteristics of the human visual system (HVS), to effectively assess the quality of underwater images. Inspired by the absorption and scattering properties of underwater environments, Wang et al. [21] proposed a NR-UIQA method CCF for color images, which measures color features, contrast features, and fog density, and combines these features linearly to characterize image quality. Yang et al. [22] designed a frequency-domain NR-UIQA metric FDUM, which combines color, dark channel prior-weighted contrast measurement, and sharpness, synthesizing these measurements into a comprehensive quality assessment using weights determined by multiple linear regression. Jiang et al. [23] established a new underwater image enhancement (UIE) dataset SAUD, and proposed a color space-based NR-UIQA metric. Li et al. [24] constructed an underwater image quality assessment dataset USRD, which includes synthetic and real-world underwater images, and proposed an NR-UIQA based on the transmission diagram of underwater imaging. Yi et al. [25] proposed a noticeable NR-IQA method by analyzing the colorfulness, contrast, and visibility characteristics of underwater enhanced images. Liu et al. [26] proposed an NR-UIQA, which describes image quality by integrating various features such as brightness, color cast, sharpness, contrast, and fog density. This model uses SVR to establish a model that maps the relationship between features and image quality. Hou et al. [27] proposed an NR-UIQA method to predict underwater image quality by extracting features such as contrast, sharpness, and naturalness, and using SVR to establish a model between features and subjective quality. Zhang et. al. [28] proposed a NR-UIQA method based on quality-aware features, which extracted features such as naturalness, color, contrast, clarity, and structure, and trained the model through Gaussian process regression to predict image quality. Jiang et al. [29] established a multi-dimensional quality annotation dataset SAUD2.0, applied to underwater enhanced images, and proposed a quality evaluation method based on a multi-stream collaborative learning network. Liu et al. [30] constructed a database of real underwater images and built a NR-UIQA based on a deep learning framework that incorporates channel and spatial attention mechanisms along with a transformer module.

## 3. The Proposed MFEM-UIQA

The proposed framework for MFEM-UIQA is shown in Figure 1. In order to evaluate the quality of underwater images, the chrominance features, luminance features, and perception features of underwater images are extracted, respectively, and further combined into feature vectors. In the training stage, SVR is used to build a predictive model of underwater images, which is used to describe the mapping relationship between underwater image features and image quality. During the testing phase, features are extracted from the test images. Subsequently, the trained prediction model utilizes these features to derive the prediction quality scores. The following chapters will detail how to effectively extract these features.

### 3.1. Underwater Image Chrominance Features

#### 3.1.1. Chromatic Statistical Feature

Research has revealed that light undergoes varying degrees of attenuation when propagating through water, with the red wavelengths experiencing the most pronounced decay, while blue and green light waves are relatively more preserved [32]. This phenomenon imparts a blue or green hue to underwater colors, rendering the color cast in underwater imagery particularly conspicuous. Studies in human vision have indicated that color perception typically occurs within the framework of the OC Space. Consequently, we initially shift the image color representation from RGB to OC, and subsequently extract color cast information by calculating the differences in chrominance components. The OC space comprises three color components denoted as *Ψ*_1_, *Ψ*_2_, and *Ψ*_3_. *Ψ*_1_,*Ψ*_2_, and *Ψ*_3_ denote the red-green, yellow-blue chrominance, and luminance components separately; $\Psi_{1}=(R-G)/\sqrt{2}$, $\Psi_{2}=(R+G-2B)/\sqrt{6}$, $\Psi_{3}=\left(R+G+B\right)/\sqrt{3}$. Subsequently, the definition of the UCD map *Ψ_D_* based on the OC space is as follows:(1)$$\Psi_{D}=\left|\Psi_{1}-\Psi_{2}\right|$$

The original underwater image with different Mean Opinion Scores (MOSs) and corresponding UCD maps are illustrated in Figure 2. It can be seen that the derived UCD maps exhibit a high degree of correlation within local regions.

The UCD map *Ψ_D_* reflects the color cast of underwater images, so that statistical features can be extracted from *Ψ_D_*. The following stage in preprocessing involves the application of local contrast normalization to mimic the non-linear masking properties inherent in human visual perception, which is utilized to produce Mean Subtracted Contrast Normalized (MSCN) images *Ψ_D_* [33]:(2)$${\hat{\Psi }}_{D}(i,j)=\frac{\psi_{D}(i,j)-\mu_{\Psi_{D}}(i,j)}{\sigma_{\Psi_{D}}(i,j)+c}$$
where $\psi_{D}(i,j)$ represents the pixel values at the coordinate position (i,j) of the chromatic difference map, while c serves as a constant to prevent division by zero, μ denotes the mean, and σ denotes the variance, which are defined as follows:(3)$$\mu_{\Psi_{D}}(i,j)=\sum_{s=-S}^{S}\sum_{n=-T}^{T}\omega_{s,t}\psi_{D}(i,j)$$(4)$$\sigma_{\Psi_{D}}(i,j)={\left[\sum_{s=-S}^{S}\sum_{t=-T}^{T}\omega_{s,t}{\left[\psi_{D}(i,j)-\mu (i,j)\right]}^{2}\right]}^{\frac{1}{2}}$$
where (*i*,*j*) represent the pixel-level coordinate values, i∈[0,M] j∈[0,N], while *M* and *N* represent the height and width of the image, ω_s,t_ is defined as a center-symmetric Gaussian weighted filtering window, given by $\omega =\left\{\omega_{s,t}|m=\left[-S,\ldots ,S\right],n=\left[-T,\ldots ,T\right]\right\}$, and S and T are set to three [33].

Figure 3 shows the statistical distribution of MSCN coefficients maps for the four underwater original images (Figure 2a) and corresponding *Ψ_D_* (Figure 2b). It is observable in Figure 3a that, as the degree of distortion in images varies, the histogram distributions of the MSCN coefficients of the original underwater images do not show a consistent pattern of change that correlates with the level of distortion. In contrast, Figure 3b can clearly distinguish different image qualities, that is, with the increase in image distortion, the statistical graph variance of the MSCN coefficient of *Ψ_D_* presents an obvious rule. Therefore, the coefficients of the MSCN statistical distribution of *Ψ_D_* can reflect the change in underwater image quality more effectively than the original image.

Owing to precision in detecting minor statistical fluctuations [34], the Asymmetric Generalized Gaussian Distribution (AGGD) is adopted to fit the MSCN coefficient map ${\hat{\Psi }}_{D}$. The mathematical formulation of AGGD is detailed subsequently [35]:(5)$$f_{\Psi_{D}}({\hat{\psi }}_{d},\sigma_{l}^{2},\sigma_{r}^{2},\gamma )=\left\{\begin{array}{l} \frac{\gamma }{\left(\beta_{l}+\beta_{r})\Gamma (1/v\right)}\exp(-{\left(-{\hat{\psi }}_{d}/\beta_{l}\right)}^{\gamma }),{\hat{\psi }}_{d}<0\\ \frac{\gamma }{\left(\beta_{l}+\beta_{r})\Gamma (1/v\right)}\exp(-{\left(-{\hat{\psi }}_{d}/\beta_{r}\right)}^{\gamma }),{\hat{\psi }}_{d}\geq 0 \end{array}\right.$$
where $\beta_{l}=\sigma_{l}\sqrt{\frac{\Gamma (1/\alpha )}{\Gamma (3/\alpha )}}$, $\beta_{r}=\sigma_{r}\sqrt{\frac{\Gamma (1/\alpha )}{\Gamma (3/\alpha )}}$, $\eta =\left(\beta_{r}-\beta_{l}\right)\sqrt{\frac{\Gamma (2/v)}{\Gamma (1/v)}}$, and AGGD is characterized by four parameters: γ, η, $\sigma_{l}^{2}$ and $\sigma_{r}^{2}$, respectively. γ modulates the shape of the fitting distribution, whereas parameters $\sigma_{l}$ and $\sigma_{l}$ are responsible for modifying the dispersion of the AGGD symmetrically around its central value. The parameter η is designated to signify the mean for achieving the best fit of the AGGD model. Consequently, the parameters $\left[\sigma_{l},\sigma_{r},\gamma ,\eta \right]$ derived from the AGGD fitting are utilized as descriptors for statistical chromatic characteristics.

#### 3.1.2. Rayleigh Distribution Feature

Additionally, when assessing image quality, the Probability Density Function (PDF) offers a robust statistical approach at the image level. Notably, the Rayleigh distribution has gained widespread acceptance due to its close alignment with the intensity histograms of underwater images [36]. In this study, the PDF of the Rayleigh distribution is utilized to characterize the intensity profiles of underwater images as follows:(6)$$f_{R}(x)=\frac{x}{\sigma_{\Psi i}^{2}}\exp(-\frac{x^{2}}{2\sigma_{\Psi i}^{2}}),\;\;\;\;\;\;\;\;x>0,$$
where $\sigma_{\Psi_{i}}$ represents the shape parameter of the Rayleigh distribution in the image.

Figure 4 illustrates the relationship between the quality grades of underwater images and the shape parameter of the Rayleigh distribution of the three channels in the OC space. It can be observed that there is a positive correlation between the image quality and the shape parameter values. That is, as the image quality improves, the corresponding shape parameter values increase. Consequently, the shape parameter of the Rayleigh distribution can serve as an indicator of the colorfulness of underwater images. Therefore, the shape parameters $\left[\sigma_{\Psi_{1}},\sigma_{\Psi_{2}},\sigma_{\Psi_{3}}\right]$ of the three components of OC space are incorporated into the chrominance features to enhance the description of image quality.

#### 3.1.3. Multi-Channel MI Based on Entropy

Additionally, image entropy is an important indicator for measuring the distortion and statistical characteristics of image information. Its capability to effectively capture the degree of information variation aids in gaining a deeper understanding of the overall information richness and structural complexity of an image. We first extract one-dimensional and two-dimensional entropies from the OC [*Ψ*_1_, *Ψ*_2_, *Ψ*_3_] space images and subsequently calculate the MI features based on image entropy, thereby reflecting the multi-channel entropy characteristics of the image.

Take channels *Ψ*_1_ and *Ψ*_2_ for example, and the definitions of one-dimensional entropy are as follows:(7)$$\left\{\begin{array}{l} E_{1}(\Psi_{1})=-\sum_{\psi_{1}=0}^{255}p(\psi_{1})\log_{2}p(\psi_{1})\\ E_{1}(\Psi_{2})=-\sum_{\psi_{2}=0}^{255}p(\psi_{2})\log_{2}p(\psi_{2}) \end{array}\right.$$
where $\psi_{1}$ and $\psi_{2}$ represent the pixel values of the respective OC images, while $p(\psi_{1})$ and $p(\psi_{2})$ denote the probabilistic distribution functions for each of these channels.

Furthermore, the two-dimensional entropy is defined as follows [37]:(8)$$E_{2}(\Psi_{1},\Psi_{2})=-\sum_{\psi_{1}=0}^{255}\sum_{\psi_{2}=0}^{255}p(\psi_{1},\psi_{2})\log_{2}p(\psi_{1},\psi_{2})$$
where $p(\psi_{1},\psi_{2})$ is the joint probabilistic distribution function.

Therefore, the mutual information entropy between the ***Ψ*_1_** and ***Ψ*_2_** channels can be calculated as follows:(9)$$MI_{E}(\Psi_{1},\Psi_{2})=E_{1}(\Psi_{1})+E_{1}(\Psi_{2})-E_{2}(\Psi_{1},\Psi_{2})$$
where $E_{1}(\Psi_{1})$ and $E_{1}(\Psi_{2})$ represent the one-dimensional entropies of images *Ψ*_1_ and *Ψ*_2_, respectively. $E_{2}(\Psi_{1},\Psi_{2})$ denotes the two-dimensional entropy of these two images. Finally, the entropy-based MI features of the [*Ψ*_1_, *Ψ*_2_, *Ψ*_3_] color channels are extracted to further characterize the chromaticity features of the underwater image.

### 3.2. Underwater Image Luminance Features

#### 3.2.1. Contrast Feature

Research has indicated that photoreceptor cells in the retina, specifically the cones and rods, exhibit a high degree of responsiveness to variations in light intensity [38]. The contrast feature effectively reflects the sensitivity of the human eye to luminance differences, thereby accurately capturing the visibility of details and structures within an image and more closely emulating the HVS perceptual characteristics. The contrast feature $F_{GC}$ is calculated as follows [39]:(10)$$F_{GC}={\sum_{k=1}^{K}\omega_{k}C}_{k}$$
where ***ω_k_*** represents the weight factor, ***C_k_*** denotes the contrast coefficient, and K represents the scaling levels of different magnitudes within an image, which is set to nine [40]. The specific definitions of ***ω_k_*** and ***C_k_*** are as follows:(11)$$\omega_{k}=\alpha \times {\left(\frac{k}{K}\right)}^{2}+\beta \times \frac{k}{K}+\gamma$$(12)$$C_{k}=\frac{1}{M\times N}\sum_{i=1}^{M}\sum_{j=1}^{N}l_{C(i,j)}$$
where *M* denotes the height of the image, while *N* corresponds to the width of the image.$l_{C(i,j)}$ quantifies the average luminance disparity between a given pixel and its surrounding pixels within an image matrix, which is specifically defined as follows [40]:(13)$$l_{C(i,j)}=\left(\left|\gamma_{\left(i,j\right)}-\gamma_{\left(i,j-w\right)}\right|+\left|\gamma_{\left(i,j\right)}-\gamma_{\left(i-1,j\right)}\right|+\left|\gamma_{\left(i,j\right)}-\gamma_{\left(i,j+w\right)}\right|+\left|\gamma_{\left(i,j\right)}-\gamma_{\left(i+1,j\right)}\right|\right)/4$$
where $\gamma_{\left(i,j\right)}$ represents the pixel value after undergoing gamma correction, with a gamma value set to 2.2, which simulates the non-linear response of human vision to brightness. By calculating the contrast coefficient, one can assess the overall contrast level of the image. Finally, the contrast feature $F_{GC}$ is obtained to characterize the luminance features of the image.

#### 3.2.2. Uniformity Feature

Secondly, from the vantage point of visual physiology, the sensitivity of rod cells to luminance variations and the role of cone cells in color perception jointly dictate the quality and precision of visual signals. An imbalance in luminance distribution can disrupt the responses of these photoreceptor cells, thereby inducing biases in the conveyance and processing of visual information [41]. Such non-uniformity not only pertains to the overall luminance characteristics of an image but is also a pivotal determinant in the assessment of image quality. For example, as depicted in Figure 5a, an underwater image exhibiting uneven luminance features has a MOS value of merely 0.4, underscoring the adverse impact of luminance non-uniformity on image quality.

We present a more concise computational method to quantify the luminance uniformity of underwater images. Initially, the OC space luminance component image Ψ_3_ is divided into *N* × *N* non-overlapping blocks. Subsequently, the average luminance of each block is calculated. The luminance uniformity coefficient is then defined as follows:(14)$$F_{Hum}=\frac{{\bar{I}}_{max}-{\bar{I}}_{min}}{{\bar{I}}_{max}+c}$$
where ${\bar{I}}_{max}$ denotes the block with the maximum average luminance, while ${\bar{I}}_{min}$ represents the block with the minimum average luminance. The constant c is introduced to enhance the stability of the calculation, with an experimental value of 0.00001. A larger value of $F_{Hum}$ indicates a lower level of luminance uniformity in the image, whereas a smaller value suggests better luminance uniformity. Considering complexity and practical effectiveness, the value of *N* is set to 5 in the experiments, as illustrated in Figure 5b. By calculating the average luminance values of these blocks, the luminance uniformity coefficient $F_{Hum}$ is obtained to serve as luminance features.

#### 3.2.3. Entropy-Based MI of Luminance Subbands

Thirdly, in underwater environments, the scattering and absorption of light leads to the loss of image luminance information, resulting in a decline in image quality. Since image entropy can effectively represent image information loss, a luminance MI feature based on entropy is adopted. First of all, the Fourier transform is applied to the underwater image, and then the logarithmic Gabor filter is used to extract subband images in multi-scales and directions. The logarithmic Gabor filter is defined as follows [42]:(15)$$F_{G}(f,\theta )=\exp(-\frac{{\left(\log(f/f_{0})\right)}^{2}}{2{\left(\log(\sigma_{r}/f_{0})\right)}^{2}}-\frac{{\left(\theta /\theta_{0}\right)}^{2}}{2\sigma_{\theta }^{2}})$$
where f_0_ denotes the center frequency, while θ_0_ signifies the principal orientation, and σ_r_ and σ_θ_ correspond to the radial and angular spread, respectively. By applying the aforementioned filter, the original image is decomposed into directional subband components oriented at 0°, 45°, 90°, and 135°, designated as F1, F2, F3, and *F_4_*. Furthermore, the calculations similar to Formula (9) are carried out to obtain the entropy-based MI values of luminance subbands. Finally, the MI features of luminance subbands are used to further characterize the luminance features of underwater images.

### 3.3. Underwater Image Perception Features

#### 3.3.1. Noise Feature

Research has revealed that images with different noise levels exhibit significant differences in the degree of spatial frequency degradation after the application of a fixed low-pass filter. Specifically, images with higher noise levels experience more pronounced attenuation in spatial frequency following low-pass filtering [43]. It can be inferred from this that there is a greater difference between underwater images with higher noise levels and images that have been processed with a low-pass filter. The low-pass filtering difference map is calculated as follows:(16)$$\Psi_{LPF}=\Psi_{3}-G_{LPF}$$

To accurately describe these subtle statistical changes, the Generalized Gaussian Distribution (GGD) model with zero central tendency is utilized for fitting, the mathematical expression of which is as follows [44]:(17)$$f_{\Psi_{LPF}}(\psi_{LPF};\lambda ,\sigma_{\Psi_{LPF}}^{2})=\frac{\lambda }{2\beta_{\Psi_{LPF}}\Gamma (1/\lambda )}\exp(-{\left(\left|\psi_{LPF}\right|/\beta_{\Psi_{LPF}}\right)}^{\lambda })$$
where(18)$$\beta_{\Psi_{LPF}}=\sigma_{\Psi_{LPF}}\sqrt{\frac{\Gamma (1/\lambda )}{\Gamma (3/\lambda )}}$$
where Γ() is Gamma function. GGD encompasses two parameters: the shape parameter λ and variance parameter $\sigma_{\Psi_{LPF}}^{2}$. Finally, GGD parameters $\left(\lambda ,\sigma_{\Psi_{LPF}}^{2}\right)$ are extracted to serve as noise features.

#### 3.3.2. Multi-Channel Dispersion Feature

Moreover, researchers have revealed disparities in the histogram fitting profiles of underwater images, distinguishing between those of inferior and superior quality. The Kullback–Leibler (K-L) divergence serves as an effective metric for quantifying the divergence in probabilistic density across distinct distributions [45]. In particular, let the functions *p*(*x*) and *q*(*x*) represent the probability density functions associated with the Rayleigh distribution for degraded and pristine underwater images, respectively. The divergence between these distributions is articulated as follows [45]:(19)$$D_{KL}(p||q)=\int_{-\infty }^{\infty }-p(x)[\log q(x)-\log p(x)]dx\;$$
where *p*(*x*) and *q*(*x*) represent the probability density functions of the Rayleigh distribution for images with different degrees of distortion. The histogram distribution parameters for the high-quality images are derived from [46].

The comparison of the K-L divergence between the distorted and pristine images under the Rayleigh distribution are demonstrated in Figure 6. The lower K-L divergence signifies a more proximate distribution alignment between the test image and pristine image, thereby implying a superior image quality. Conversely, The higher K-L divergence implies a poorer image quality. Taking into account the collective influence of the $\left[\Psi_{1},\Psi_{2},\Psi_{3}\right]$ channels, the multi-channel dispersion features $\left[D_{KL}^{\Psi_{1}},D_{KL}^{\Psi_{2}},D_{KL}^{\Psi_{3}}\right]$ are extracted to serve as the perception features.

#### 3.3.3. Visibility Feature

Furthermore, the forward scattering effect in underwater environments can lead to image blurring and a loss of detail, thereby reducing image visibility and subsequently affecting image quality. Consequently, image visibility features are crucial for assessing image quality [47]. To assess image visibility, we utilized the multi-channel weighted ***EME*** to assess the visibility of images. The computational formula is presented below [20]:(20)$$F_{V}=\sum_{c=1}^{3}\lambda_{c}E_{M}(I_{c})$$
where $\lambda_{c}$ denotes the visual response coefficients for the red, green, and blue channels of the RGB color space, with values of 0.299, 0.587, and 0.114, respectively [22]. These coefficients more accurately reflect the HVS perception by the human eye and the collective influence of multiple color channels. The definition of ***EME*** is as follows:(21)$$E_{M}=\frac{2}{K}\sum_{j=1}^{K}\log(\frac{I_{\max,j}}{I_{\min,j}})$$
where the image is segmented into k discrete patches, $I_{\min,j}$ and $I_{\max,j}$, which represent the minimum and maximum intensity values within each patch, respectively.

Figure 7 demonstrates a significant positive correlation between the visibility features of images and their subjective quality. Specifically, samples with a higher image quality exhibit higher visibility values. Consequently, visibility features $F_{V}$ are adopted to serve as the perception features.

### 3.4. Feature Regression

After obtaining the underwater image chrominance features, luminance features, and perceptual features, all these features are consolidated into a feature vector set $\;\vec{F}=[f_{1},\;f_{2},\;\ldots ,\;f_{30}]$. After the feature vector is obtained, the regression model is used to establish the mapping relationship from the feature to the quality. The framework is general enough that multiple regression tools can be used. In this study, the widely used SVR with the radial basis function kernel (RBSK) is adopted [48,49]. As a traditional machine learning approach, using $\vec{F}\;$ and the corresponding sample labels  Q of the training image set Ω, the model M is trained through SVR:(22)$$M=T({\vec{F}}_{i},Q_{i}),i\in \Omega$$
where i represents the index of an image within the datasets and T denotes the training function.

In the testing phase, for any given test image with feature vector $\;\vec{F^{\prime }}=[f_{1},\;f_{2},\;\ldots ,\;f_{30}]$, the trained model M could be utilized to obtain the predicted quality score:(23)$$Q^{\prime }=P(\vec{F^{\prime }},M)$$
where $\;Q^{\prime }$ represents the predicted quality value and P denotes the prediction function.

## 4. Experiment Results and Discussion

### 4.1. Experiment Protocol

To comprehensively evaluate the performance of the proposed MFEM-UIQA algorithm, three datasets USRD [24], UWIQA [22], and UID2021 [50] were selected in this experiment. The USRD dataset comprises a total of 420 artificially generated underwater images and 320 authentic underwater photographs, establishing a comprehensive underwater image resource library, corresponding with the MOS of each image. The UWIQA dataset comprises 890 underwater images from real-world environments, which not only bolsters the verisimilitude and variegation of the dataset but also bolsters the dataset’s genuine reliability. The UID2021 dataset encompasses a curated collection of 900 underwater images, refined via a multitude of methodologies. This dataset is derived from an extensive enhancement process applied to 60 distinct original underwater images, which incorporate intricate elements and have been subjected to 15 diverse UIE algorithms. Compared with the three datasets, the USRD dataset consists of underwater natural images and synthetic images, and contains a variety of scenes. The UWIQA dataset has more underwater images, more abundant scenes, and all of them are natural and real underwater images, which can test the algorithm more comprehensively. The UID2021 dataset is composed of original underwater images and corresponding underwater enhanced images, which can verify the generalization performance of the proposed algorithm for UIE tasks. The use and testing of these three datasets can evaluate the performance of the algorithm more comprehensively from many aspects. With the application of these three datasets, the performance of the algorithm can be evaluated more comprehensively.

Furthermore, a series of IQA models is adopted to compare with the MFEM-UIQA. This includes widely utilized NR-IQA algorithms, such as BRISQUE [8], NIQE [9], IL-NIQE [10], and BPRI [11], as well as NR-IQA methods specifically tailored for fog and contrast degradation, including DBCP [14], FADE [51], and VDA-DQA [15]. Additionally, this research considered no-reference IQA models designed for underwater environments, comprising UCIQE [19], UIQM [20], CCF [21], and FDUW [22]. All of these models have been tested and evaluated in order to validate the performance and applicability of MFEM-UIQA.

### 4.2. The Experimental Results on Each Dataset

Furthermore, three metrics are employed to assess the performance of IQA methods, the Pearson Linear Correlation Coefficient (PLCC), the Spearman Rank Order Correlation Coefficient (SROCC), and the Root Mean Square Error (RMSE). These quantitative criteria are designed to precisely evaluate the congruence between the estimated image quality metrics and the empirical MOS data. Optimally, the PLCC and SROCC are expected to converge towards a value of 1, indicating a perfect correlation, whereas the RMSE aims to minimize its deviation, ideally approaching zero.

The experimental procedures were executed on a computing environment, outfitted with an Intel(R) Core (TM) i5-7200 CPU operating at 2.10 GHz and furnished with 16 GB of RAM. The computations relied on MATLAB 2022b. In the regression experiment, we adopted the widely used SVR [48], which has been used in many IQA methods [8,11,13,26,27]. The LIBSVM library was deployed to implement the SVR regression. During the experiment, we used the default parameters, except hyperparameters **γ** and the regularization parameter **C** of RBSK [49]. Before the formal training, we first performed several pre-trainings, from which the optimal **γ** and **C** were selected, and then the formal training was carried out. In order to reduce bias, the dataset was divided into 80% for training and 20% for testing randomly in the formal training process. This process was iterated 1000 times. The conclusive assessments were the median values of the metrics from these iterations.

Table 1 demonstrates that the proposed MFEM-UIQA algorithm outperforms other IQA algorithms on the USRD database, with the highest PLCC and SROCC values and the lowest RMSE value. Specifically, the PLCC is 0.8919, the SROCC is 0.8881, and the RMSE is 0.1101, indicating a high correlation with human visual perception. Compared to other general NR-IQA algorithms, MFEM-UIQA improved upon the best-performing BPRI by 16.2% in PLCC and by 21.6% in SROCC. When compared to IQA algorithms specifically designed for fog density and defogging, MFEM-UIQA enhanced the PLCC by 42.5% and the SROCC by 45.1% over VDA-DQA. In comparison with IQA algorithms specialized for underwater images, MFEM-UIQA improved the PLCC by 11.6% and the SROCC by 14.9% over the best-performing FDUW. Therefore, the proposed MFEM-UIQA exhibits the best performance over methods in the assessment of underwater image quality.

To further validate the performance of MFEM-UIQA on real underwater image assessment, we evaluated various NR-IQA methods on the UWIQA dataset, with the results presented in Table 2. Compared to other algorithms, the proposed MFEM-UIQA demonstrated a superior performance, achieving the highest PLCC and SROCC values of 0.7679 and 0.7644, respectively, indicating the highest consistency between its predicted quality scores and actual subjective scores. Additionally, it exhibited the lowest RMSE value of 0.1064, indicating the smallest prediction error among the compared algorithms.

Currently, the UIE techniques are dedicated to improving underwater image quality. However, the enhancement process may lead to over or under enhancement, affecting the performance in practical applications. We applied these techniques to evaluate the performance of the MFEM-UIQA algorithm, for validating its applicability in a more diverse range of environments. The results of the testing of various algorithms on the UID2021 dataset are presented in Table 3. Among various approaches, the MFEM-UIQA algorithm demonstrated a superior performance, achieving a PLCC of 0.7170 and SROCC of 0.7052. Therefore, the proposed MFEM-UIQA is more effective for accessing the quality of UIE images.

### 4.3. Ablation Experiment

To assess the contribution of each module to the overall performance of the algorithm, we conducted an ablation experiment on the USRD database. The following lists the various feature module combinations:M1: Full features;M2: Underwater image chrominance features;M3: Underwater image luminance features;M4: Perception features;M5: Underwater image chrominance and perception features;M6: Underwater image luminance and perception features.

As demonstrated in Table 4, the results of the ablation experiment indicate that each feature group contributes to the overall performance, with a significant enhancement observed when multiple features are integrated. This suggests that all the features are important for the proposed MFEM-UIQA. Taking the PLCC metric as an example, the performance of the chrominance feature M2 is 0.8017, the performance of the luminance feature M3 is 0.8215, and the performance metric of the perceptual feature M4 is 0.7804. Notably, when module features are integrated, the performance is significantly improved. The combination of chrominance features and perception features, M5, achieves a performance metric of 0.8725; the combination of luminance and perceptual features M6 achieves a performance metric of 0.8588. It is evident that the performance of the algorithm after module feature integration is superior to the effect of individual modules. Considering the combined impact of common distortions in underwater images on chrominance, luminance, and visual perception, the result is reasonable. In particular, the combination M1, which integrates all features, demonstrates the best performance of 0.8919 among all the feature combinations. The integration of all the features outperforms individual feature components, thereby validating the effectiveness of multi-feature fusion in MFEM-UIQA.

### 4.4. T-Test Statistical Assessment

In order to substantiate the statistical relevance of the comparative performance data depicted in Table 5, a hypothesis testing framework was applied, specifically leveraging paired t-tests. The preliminary phase of this analytical procedure involved assessing the normalcy of the PLCC residuals across different IQA assessments. Should the PLCC residuals exhibit a normal distribution, a paired t-test is deemed appropriate; in cases of non-normality, a non-parametric test is considered. The null hypothesis posits that there is no difference in the mean correlations between the two UIQA assessments, while the alternative hypothesis suggests a disparity, indicating that one UIQA approach yields superior mean correlations over its counterpart. The tabulated findings in Table 5 delineate the statistical dominance: a ’1’ signifies that the row-associated method outperforms the column-associated method statistically, ’−1’ signifies the opposite, and ’0’ implies no significant statistical divergence between the IQA metrics of the row and column. Synthesizing the data from Table 5, it is inferred that MFEM-UIQA exhibits a statistically significant advantage in comparison to other IQA methodologies.

### 4.5. Dataset Partitioning Experiment

To provide a comprehensive evaluation of the proposed MFEM-UIQA, a series of experiments were designed to evaluate its performance across a spectrum of training set sizes. The proportion of the dataset allocated to training was systematically varied from 20% (with the remaining 80% for testing) up to 80% (for training, leaving 20% for testing), increasing in 10% increments, with each partition ensuring the absence of any common elements between the training and testing subsets within the USRD dataset. Table 6 illustrates the algorithm’s performance metrics PLCC, SROCC, and RMSE under these varying dataset partitions. In this context, ΔPLCC represents the improvement in PLCC relative to the preceding partition ratio.

The results reveal a positive correlation between the enlargement of the training dataset proportion and the enhancement of the MFEM-UIQA algorithm’s performance, with improvements capped at 0.0138. It is noteworthy that once the training dataset comprises 40% of the overall dataset, the PLCC increment falls beneath 0.01, indicating a stabilization in the algorithm’s efficacy. Specifically, with a mere 20% of the dataset allocated to training, the MFEM-UIQA achieved a PLCC value of 0.8455, outperforming the other algorithms as documented in Table 1. This outcome further substantiates the robustness and preeminence of the proposed approach.

### 4.6. Comparison of Different Regression Methods

In order to compare the effects of different regression methods on the performance of the algorithm, we further conduct the comparison experiment using SVR and random forest (RF) regression [52] on the USRD dataset. It can be seen from Table 7 that the effect of SVR and RF regression is continuously improved with the continuous increase in the proportion of training set partition. In the case of the same proportion of datasets, the regression effect of SVR is better than that of RF regression, indicating that the prediction effect of the model obtained by SVR regression is better. When the dataset is divided into 80% for training and 20% for testing, even if RF regression is adopted, the PLCC and SROCC still reach 0.8590 and 0.8387, which is still better than other comparison algorithms in Table 1 and achieves a good performance.

### 4.7. Computational Performance Analysis

In addition, an exemplary user-experience IQA algorithm must ensure a high-quality evaluative performance while reducing the runtime. To this end, this study selected the image set from the USRD database to measure the execution time of a series of IQA algorithms, thereby assessing the complexity of the algorithms. During the temporal efficiency assessment, it was observed that the duration allocated to feature extraction markedly surpasses that of the prediction phase, with the latter being negligible in terms of time cost. Table 8 delineates the mean computational durations for an array of IQA algorithms across the USRD dataset. Our analysis revealed that the MFEM-UIQA algorithm, as proposed, exhibits substantially reduced computational times when contrasted with other methodologies including IL-NIQE, DBCP, VDA-DQA, FADE, UIQM, CCF, and FDUM. While the computational time for MFEM-UIQA is marginally greater than for algorithms such as BRISQUE, NIQE, BPRI, and UCIQE, the performance of MFEM-UIQA is superior to these methods. Consequently, MFEM-UIQA demonstrates a balance between superior image quality assessment performance and manageable computational requirements.

## 5. Conclusions

A novel NR-UIQA method MFEM-UIQA is proposed in this paper. Considering the color shifts of underwater images, UCD maps for underwater images are created and statistical features are extracted. Moreover, the parameters of the Rayleigh distribution and entropy-based multi-channel MI features are extracted. For the luminance features, the process begins by extracting image contrast features based on gamma correction. Subsequently, acknowledging the non-uniformity inherent in underwater imaging, the luminance uniformity features are extracted. Additionally, logarithmic Gabor filtering is applied to the luminance images for subband decomposition, and entropy-based MI features are captured. Furthermore, the underwater image noise feature, multi-channel dispersion information, and visibility features are extracted to jointly represent the perceptual features. Compared with state-of-the-art methods, the proposed MFEM-UIQA shows the best performance. In the future, we will continue to deeply study the mechanisms of underwater image formation and visual perception to further improve the performance of image quality evaluation.

## Figures and Tables

**Figure 1 entropy-27-00173-f001:**
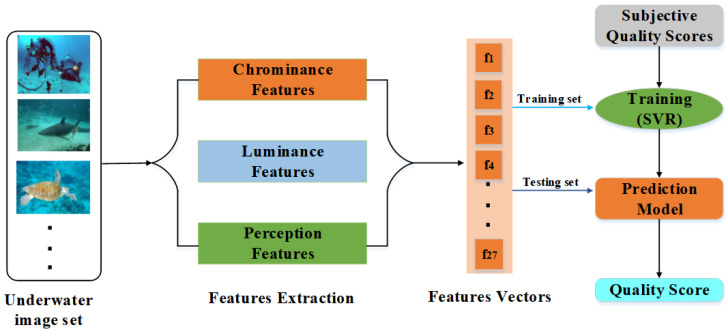
The Framework of MFEM-UIQA.

**Figure 2 entropy-27-00173-f002:**
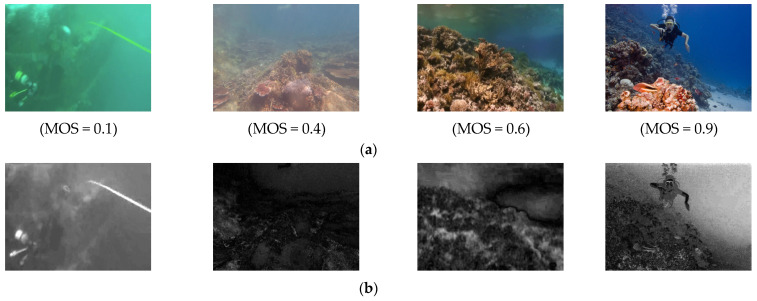
Underwater images and corresponding UCD maps. (**a**) Underwater images of different quality levels; (**b**) corresponding UCD maps.

**Figure 3 entropy-27-00173-f003:**
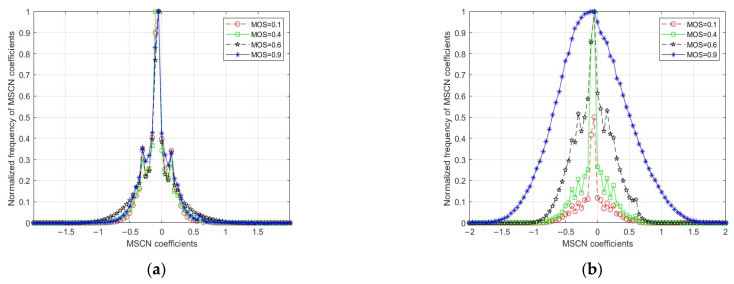
Comparison of statistical distribution of MSCN coefficients for original underwater images and corresponding *Ψ_D_*. (**a**) The statistical distribution of MSCN coefficients for the original underwater images, and (**b**) the statistical distribution of MSCN coefficients for *Ψ_D_*.

**Figure 4 entropy-27-00173-f004:**
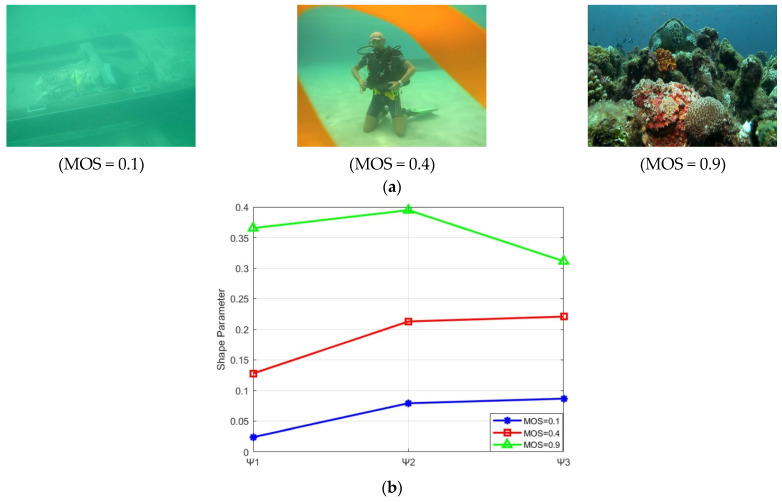
Underwater images of different quality levels and the corresponding fitting Rayleigh distribution shape parameter. (**a**) Underwater images of different quality levels; (**b**) fitting Rayleigh distribution shape parameters corresponding to three channel histograms of the OC space.

**Figure 5 entropy-27-00173-f005:**
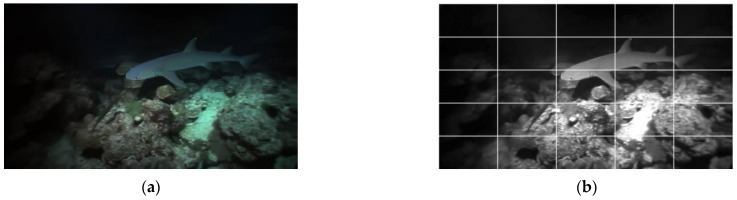
Non-uniform brightness image and its block map. (**a**) Non-uniform brightness underwater image; (**b**) block map of (**a**).

**Figure 6 entropy-27-00173-f006:**
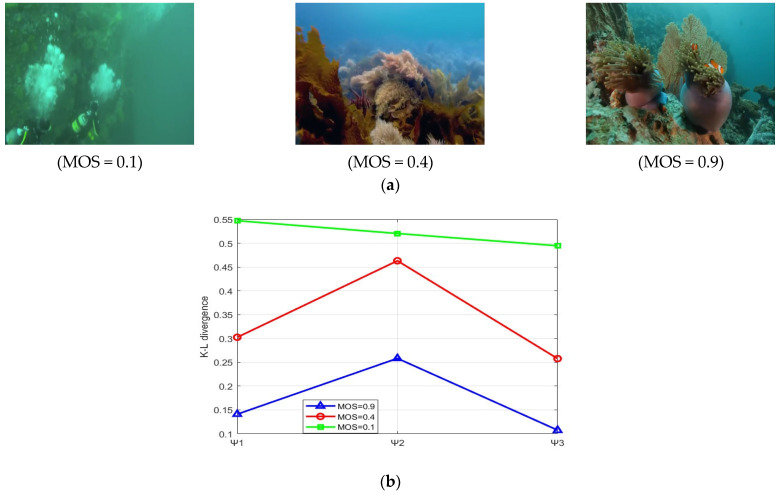
Underwater images with differing quality and corresponding K-L divergence distribution. (**a**) Underwater images of different quality levels; (**b**) the K-L divergence distribution of three channels in the OC space.

**Figure 7 entropy-27-00173-f007:**
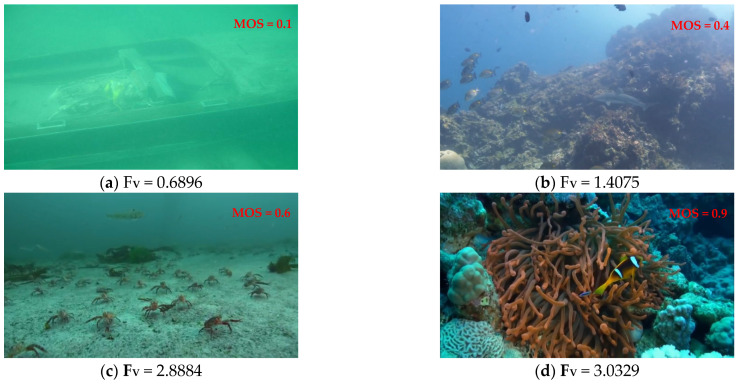
Different quality underwater images and corresponding visibility values.

**Table 1 entropy-27-00173-t001:** Experimental Results of Various Algorithms on the USRD Dataset.

Method	PLCC	SROCC	RMSE
BRSIQUE [8]	0.6016	0.5412	0.1809
NIQE [9]	0.6865	0.6432	0.1650
ILNIQE [10]	0.7343	0.7126	0.1538
BPRI [11]	0.7674	0.7304	0.1450
DBCP [14]	0.3027	0.2946	0.2149
FADE [51]	0.5247	0.5021	0.1925
VDA-DQA [15]	0.6259	0.6122	0.1770
UCIQE [19]	0.7503	0.7258	0.1497
UIQM [20]	0.6669	0.6464	1.1688
CCF [21]	0.1710	0.3334	0.2236
FDUW [22]	0.7989	0.7731	0.1365
MFEM-UIQA	0.8919	0.8881	0.1101

**Table 2 entropy-27-00173-t002:** Experimental Results of Various Algorithms on the UWIQA dataset.

Method	PLCC	SROCC	RMSE
BRSIQUE [8]	0.3880	0.3849	0.1420
NIQE [9]	0.4596	0.4362	0.1371
ILNIQE [10]	0.4460	0.4736	0.1364
BPRI [11]	0.5307	0.4793	0.1285
DBCP [14]	0.2765	0.2248	0.1470
FADE [51]	0.4153	0.4040	0.1387
VDA-DQA [15]	0.4981	0.5294	0.1316
UCIQE [19]	0.5743	0.5844	0.1246
UIQM [20]	0.5878	0.5959	0.1231
CCF [21]	0.2517	0.2914	0.1468
FDUW [22]	0.6411	0.6711	0.1182
MFEM-UIQA	0.7679	0.7644	0.1064

**Table 3 entropy-27-00173-t003:** Experimental Results of Various Algorithms on the UID2021 dataset.

Method	PLCC	SROCC	RMSE
BRSIQUE [8]	0.2604	0.3201	2.0047
NIQE [9]	0.3389	0.3313	2.0467
ILNIQE [10]	0.4635	0.4624	1.9114
BPRI [11]	0.4543	0.4224	1.9206
DBCP [14]	0.2747	0.1730	2.0758
FADE [51]	0.3560	0.3435	2.0176
VDA-DQA [15]	0.5723	0.5653	1.7692
UCIQE [19]	0.6255	0.6020	1.6963
UIQM [20]	0.5817	0.5400	1.7683
CCF [21]	0.4712	0.4356	1.9154
FDUW [22]	0.6372	0.6261	1.6751
MFEM-UIQA	0.7170	0.7052	1.5006

**Table 4 entropy-27-00173-t004:** Performance of Different Feature Combinations.

	PLCC	SROCC	RMSE
M1	0.8919	0.8881	0.1101
M2	0.8017	0.7733	0.1348
M3	0.8215	0.7984	0.1286
M4	0.7804	0.7563	0.1412
M5	0.8725	0.8555	0.1096
M6	0.8588	0.8417	0.1155

**Table 5 entropy-27-00173-t005:** Paired T-test Analysis of PLCC Scores for Diverse IQA methods on the USRD Dataset. G1-G12 correspond to the following IQA algorithms: BRISQUE [8], NIQE [9], IL-NIQE [10], BPRI [11], DBCP [14], FADE [51], VDA-DQA [15], UCIQE [19], UIQM [20], CCF [21], FDUW [22], and MFEM-UIQA.

Method	G1	G2	G3	G4	G5	G6	G7	G8	G9	G10	G11	G12
G1	0	−1	−1	−1	1	1	−1	−1	−1	1	−1	−1
G2	1	0	−1	−1	1	1	1	−1	1	1	−1	−1
G3	1	1	0	−1	1	1	1	−1	1	1	−1	−1
G4	1	1	1	0	1	1	1	1	1	1	−1	−1
G5	−1	−1	−1	−1	0	−1	−1	−1	−1	1	−1	−1
G6	−1	−1	−1	−1	1	0	−1	−1	−1	1	−1	−1
G7	1	−1	−1	−1	1	1	0	−1	−1	1	−1	−1
G8	1	1	1	−1	1	1	1	0	1	1	−1	−1
G9	1	−1	−1	−1	1	1	1	−1	0	1	−1	−1
G10	−1	−1	−1	−1	−1	−1	−1	−1	−1	0	−1	−1
G11	1	1	1	1	1	1	1	1	1	1	0	−1
G12	1	1	1	1	1	1	1	1	1	1	1	0

**Table 6 entropy-27-00173-t006:** Performance of Varying Train-Test Splits on USRD Dataset.

Train-Test	PLCC	SROCC	RMSE	ΔPLCC
20–80%	0.8455	0.8297	0.1208	
30–70%	0.8594	0.8455	0.1157	0.0138
40–60%	0.8674	0.8548	0.1124	0.0080
50–50%	0.8712	0.8588	0.1109	0.0038
60–40%	0.8764	0.8640	0.1087	0.0052
70–30%	0.8838	0.8698	0.1065	0.0073
80–20%	0.8919	0.8881	0.1101	0.0081

**Table 7 entropy-27-00173-t007:** Performance Comparison of Different Regression Methods on USRD Dataset.

Train-Test	SVR			RF		
	PLCC	SROCC	RMSE	PLCC	SROCC	RMSE
20–80%	0.8455	0.8297	0.1208	0.8190	0.7983	0.1299
50–50%	0.8712	0.8588	0.1109	0.8450	0.8276	0.1210
80–20%	0.8919	0.8881	0.1101	0.8590	0.8387	0.1154

**Table 8 entropy-27-00173-t008:** The Average Running Time of the Proposed MFEM-UIQA and the other IQA Methods.

Metric	BRISQUE [8]	NIQE [9]	IL-NIQE [10]	BPRI [11]	DBCP [14]	FADE [51]
Running times (s)	0.7423	0.6915	17.1417	0.8338	10.7856	8.4024
Metric	VDA-DQA [15]	UCIQE [19]	UIQM [20]	CCF [21]	FDUW [22]	MFEM-UIQA
Running times (s)	15.4775	0.4211	14.0241	13.1216	20.2020	5.3209

## Data Availability

The original contributions presented in this study are included in the article. Further inquiries can be directed to the corresponding author.

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
