# Peer review of "Multi-Space Feature Fusion and Entropy-Based Metrics for Underwater Image Quality Assessment"

_entropy, 2025, doi:10.3390/e27020173_

Round 1
Reviewer 1 Report
Comments and Suggestions for Authors
In this paper, the authors propose a new metric, namely MFEM-UIQA, for evaluating underwater image quality. Compared to serval UIQA methods, the proposed MFEM-UIQA demonstrates a superior performance. The comments are as follows.
1. At the end of the section “introduction”, the description and the contributions of the proposed method needs to be refined. Also the authors needs to better point out the difference between the proposed method and some other SVR-based underwater IQA metrics.
2. In the subsection “2.2. Underwater-Specific IQA”, some latest work is suggested to be reported (e.g., Hou et. al., “No-reference quality assessment for underwater images, 2024”, Zhang et. al., “A no-reference underwater image quality evaluator via quality-aware features”).
3. How to define “Perception Features”, what’s the difference from the first two features.
4. As is well known, due to the absorption and scattering of light in water, the image appears to have different degrees of color distortion. Therefore, color is an important comment for evaluating underwater image quality, why the proposed method did not consider the color index?
5. It is suggested to give some descriptions about the difference of the three datasets used in the experiments.
6. The English writing can be improved and polished. Also, the authors are encouraged to carefully proofread this paper and correct all syntax error.
7. The layout of this paper needs to be improved.
Comments on the Quality of English LanguageThe English writing can be improved and polished. Also, the authors are encouraged to carefully proofread this paper and correct all syntax error.
Reviewer 2 Report
Comments and Suggestions for Authors
Line 80: "spatial" - in what sense? Pixel space, chroma space, etc.
Line 81: How is the chrominance difference defined? Which color space is being used? In L*a*b space L carries luminance information and a and b - chroma information. Although these details are described later in the beginning, they do cause confusion.
Figure 2: "MOS" is defined later than used in captions.
Eqn.3: One should assume that I is a pixel value?
Eqn.3: If M and N are image(s) dimensions, why the summation goes from -M to +M (same for N)?
Figure 3: Images certainly have very different dynamic range. How do luminosity histograms for these images look like? Would they not be different?
Line 277: incorrect subscript
Line 287: problem with the caption?
Eqn.19: definition of the gamma-function is not needed.
Line 400: Kullback and Leibler deserve full names, at least when mentioned first time - "K-L" is not a widely accepted abbreviation.
Line 446: "represents" and "denotes"
Line 457: images, not imagery
Reviewer 3 Report
Comments and Suggestions for Authors
This paper presents a new method for assessing quality of underwater images. It is well structured and easy to read though some minor corrections in English are necessary. Following points needs to be addressed:
1-In section 2 ("related work"), a lengthy section about haze image analysis given. It is clear that underwater images will have unique constrains compared to atmospheric images. Focus of the background study should primarily be on the other methods used for underwater images.
2-The use of abbreviation "MOS" starts with Figure 2; however, it’s meaning is not mentioned until section 3.2.2. All the abbreviations used should be described at where they first appeared.
3- A critical part of the proposed method is the selection of features used for creating feature vector. Authors need to explain how this selection has been made particularly considering underwater images. For instance, I don’t see any frequency analysis, or any feature based on frequency analysis was used. Section 3 needs to be explained as it is it reads like a bunch of features selected randomly.
4- Similarly, fundamental part of the method proposed is the final ML algorithm used. Authors employ support vector regressor though without given any explanation. Why not use random forest for instance? How SVR is tuned?
5- Table 1 lists performance against other reported techniques in literature. However, compared against algorithms developed for atmospheric images seems redundant. Perhaps authors should indicate this is done for the sake of completeness. A true comparison would be comparing with other algorithms developed for underwater images.
6- Overall paper may have merits but as it needs some improvements to publish in a journal.
Comments on the Quality of English LanguageIt is well structured and easy to read though some minor corrections in English are necessary. For instance:
Last line in the abstract " Experimental demonstrate ... " should be "Experiments demonstrate .... "
Round 2
Reviewer 1 Report
Comments and Suggestions for Authors
The authors have addressed all my concerns.